# SARS-CoV-2 Infection in Venezuelan Pediatric Patients—A Single Center Prospective Observational Study

**DOI:** 10.3390/biomedicines11051409

**Published:** 2023-05-09

**Authors:** Francis Isamarg Crespo, Soriuska José Mayora, Juan Bautista De Sanctis, Wendy Yaqueline Martínez, Mercedes Elizabeth Zabaleta-Lanz, Félix Isidro Toro, Leopoldo Humberto Deibis, Alexis Hipólito García

**Affiliations:** 1Institute of Immunology, Faculty of Medicine, Central University of Venezuela, Caracas 1040, Venezuela; drafranciscrespo@gmail.com (F.I.C.); sori_mayo@hotmail.com (S.J.M.); wendymartinez3003@gmail.com (W.Y.M.); mercedeszabaleta@hotmail.com (M.E.Z.-L.); torfelix@gmail.com (F.I.T.); leopoldo.deibis@gmail.com (L.H.D.); 2Institute of Molecular and Translational Medicine, Faculty of Medicine and Dentistry, Palacky University, 779 00 Olomouc, Czech Republic; juanbautista.desanctis@upol.cz; 3Czech Advanced Technology and Research Institute, Palacky University, 779 00 Olomouc, Czech Republic

**Keywords:** COVID-19, pediatric COVID-19, MIS-C, lymphocyte subpopulations, IFNγ, IL-6, IL-10, malnutrition, steroids

## Abstract

Several studies suggest that children infected with SARS-CoV-2 have fewer clinical manifestations than adults; when they develop symptoms, they rarely progress to severe disease. Different immunological theories have been proposed to explain this phenomenon. In September 2020, 16% of the active COVID-19 cases in Venezuela were children under 19 years. We conducted a cross-sectional study of pediatric patients’ immune response and clinical conditions with SARS-CoV-2 infection. The patients were admitted to the COVID-19 area of the emergency department of Dr José Manuel de los Ríos Children’s Hospital (2021–2022). The lymphocyte subpopulations were analyzed by flow cytometry, and IFNγ, IL-6, and IL-10 serum concentrations were quantified using commercial ELISA assays. The analysis was conducted on 72 patients aged one month to 18 years. The majority, 52.8%, had mild disease, and 30.6% of the patients were diagnosed with MIS-C. The main symptoms reported were fever, cough, and diarrhea. A correlation was found between IL-10 and IL-6 concentrations and age group, lymphocyte subpopulations and nutritional status and steroid use, and IL-6 concentrations and clinical severity. The results suggest a different immune response depending on age and nutritional status that should be considered for treating pediatric COVID-19 patients.

## 1. Introduction

Coronaviruses (CoVs) are enveloped viruses with non-segmented, positive-sense single-stranded RNA genomes characterized by distinctive crown-like spikes that protrude from the capsid [1]. Circulating human coronaviruses can be isolated in only 4–8% of all children with acute respiratory tract infections. The disease symptoms are mild unless the child is immunocompromised [2]. SARS-CoV-2 virus shares similar characteristics and immune responses to other coronaviruses [2]; children are less frequently affected by SARS-CoV-2 [2,3,4,5].

According to statistics published by the Chinese Centers for Disease Control and Prevention, only 1% of the cases reported at the pandemic’s start were in children under 10 years of age, reaching up to 12.3% in February 2020 [2]. Several hypotheses emerged about the low incidence of COVID-19 in children: they were less exposed to the virus; they were asymptomatic or had mild symptoms similar to those of a common cold; fever was the common indication for the PCR test. Recent studies have shown that in the pediatric population, COVID-19 is often asymptomatic or less severe than in adults [2,3,4,5]. The contagion occurs through respiratory droplets, mainly in contact with an adult. Prolonged exposure to high concentrations of aerosolized virus can facilitate transmission. Gastrointestinal symptoms are the primary physical disturbance recorded [3,4,5].

The pediatric population usually is coinfected with other viruses such as Adenovirus, Bocavirus, Rhinovirus, Respiratory Syncytial Virus, Influenza, or Parainfluenza [3,4]. There seems to be a cyclical pattern with seasonal outbreaks between December and May or March to November in the southern hemisphere, which could condition the immune response and COVID-19 symptoms [3,4]. SARS-CoV-2 primarily uses the complex angiotensin-converting enzyme-2 receptor (ACE2), a glycosylated transmembrane protein, to infect and invade the target cell [6,7]. The maximum expression of ACE2 is observed in the respiratory epithelium, lungs, kidneys, intestines, testis (Sertoli and Leydig cells), uterus, vagina, endothelium, and the heart, and its expression is dependent on age [6,7,8,9]. A lower ACE2 transcript and expression in the child’s organs may explain why SARS-CoV-2 infection is less prevalent in children [6,7,8,9].

In a group of patients, a rare but severe disorder associated with COVID-19 termed multisystem inflammatory syndrome (MIS) can affect children (MIS-C) and adults (MIS-A) [10]. It is characterized by dysregulated immune responses leading to endothelial dysfunction and a hyperinflammatory state. Hyperinflammation causes a capillary leak, followed by multiorgan failure. MIS-C is characterized by inflammation in multiple organs such as the brain, kidneys, heart, eyes, lungs, skin, or gastrointestinal system [10]. The most common symptoms of MIS-C are fever, gastrointestinal (abdominal pain, vomiting, and diarrhea), and mucocutaneous (conjunctivitis, strawberry tongue, skin rash, and dried-cracked lips) [11]. Nutritional status or other medical conditions are usually not considered in pediatric COVID-19 or MIS-C [10,11,12,13,14,15]. Patients with MIS-A present comorbidities such as obesity and hypertension [10,11,12,13,14]. The male gender, in MIS-a and MIS-c, is the most affected since circulating inflammatory markers, such as IL-6 and C-reactive protein, are higher than in females [10,11,12,13,14]. Fever and rash are the most common presenting symptoms in patients with MIS-A. Since the initial clinical presentation can be non-specific, mimicking acute infection, a 4–6 weeks follow-up after COVID-19 recovery is recommended for patients with the aforementioned clinical complaints [10,11,12,13,14].

The epidemiologic dynamic of SARS-CoV-2 infection in the pediatric population in Venezuela changed dramatically, according to official statistics (https://covid19.patria.org.ve/estadisticas-venezuela/) (accessed on 24 March 2023). The percentage of positive cases reported in patients under 19 years of age changed from 2.5% (third week of September 2020) to 11% in January 2021, 15% by January 2022, and 18% in October 2022 [16]. The increase in pediatric cases may be due to different viral variants and the possible reluctance of parents to vaccinate their children against the virus.

Since the pediatric hospital Dr José Manuel de los Ríos is a national centre of reference for pediatric diseases, a particular ward was created for patients with SARS-CoV-2 infection, and specific attention was given to their nutritional status and other medical conditions. The study aimed to analyze the immune response of patients admitted to the hospital and the possible influence of different parameters, age, gender, nutritional status, treatment, and medical conditions. 

## 2. Materials and Methods

The Dr José Manuel de los Ríos Children’s Hospital is a national center for pediatric patients for children with complex medical conditions and chronic pathologies (cardiopathies, nephropathies, metabolic diseases, autoimmunity). Patients included in the study were in a special hospital ward for medical follow-up.

The study was approved by the bioethics committee of the J.M de los Ríos Children’s Hospital (HJMR-25/11/2020) and the Dr Nicolás Bianco C. Institute of Immunology, Faculty of Medicine of the Central University of Venezuela, Caracas (IDI-30/11/2020). Written informed consent from parents or guardians was taken. Informed assent from adolescents was taken along with their parents or guardians.

A descriptive cross-sectional prospective case series observational field study was conducted. The study comprises two periods in which the ward was most active, March to May 2021 and January to March 2022. A total of 72 pediatric patients with SARS-CoV-2 infection were included. The patient’s ages were between 29 days old and 18 years old. All cases met clinical/epidemiological criteria defined by the WHO, and ad admission 30 (41.7%) had a positive PCR for the virus and 42 (58.3%) had a positive Antigenic rapid diagnostic test (RDT). A structured interview was conducted with parents and representatives to obtain the information in the medical records to verify the required data and new information. The clinical history included symptoms, comorbidities, nutritional diagnosis according to WHO growth patterns, use of steroids, and MIS-C diagnosis. Patients whose parents did not sign the informed consent were excluded from the study. No patient required oxygen, and there were fatal cases.

A venipuncture was performed on all patients to obtain two tubes of 2–3 mL of peripheral blood, one with EDTA as an anticoagulant and one for a serum sample. A hematologic control was performed. Flow cytometry was used to quantify T cell subsets, B cells, and NK lymphocytes. The following monoclonal antibodies from BioLegend were used: anti-CD3/Pecy5, anti-CD4/Pe, anti-CD8/FITC; anti-CD19/PeCy5, anti-CD20/Pe, anti-CD3/FITC-CD16-CD56/Pe. The commercial ELISA kits from Legend Max™ (BioLegend) IFN-γ, IL-6, and IL-10 values were used for cytokine quantification. The ELISA tests were performed as suggested by the manufacturer. According to the manufacturer, the minimum detectable concentrations are IFN-γ 5.6 pg/mL, IL-6 1.6 pg/mL, and IL-10 2 pg/mL. In the youngest group, cytokine detection was difficult to ascertain; therefore, to facilitate the analysis, the patients were organized by age group according to age, establishing four groups: infants: 29 days old to 23 months; preschoolers: 2 years old to 6 years 11 months and 29 days old; school-age children: 7 years old to 9 years 11 months and 29 days old; adolescents: 10 years old to 18 years 11 months and 29 days old.

### Statistical Analysis

The techniques of descriptive statistics were applied for the first analysis. Tables and figures of the absolute and relative frequencies were represented. The general comparisons and necessary calculations were performed with Microsoft Excel program version 16.70. The arithmetic mean and standard deviation were calculated for the continuous variables; in the case of nominal variables, their frequencies and percentages were calculated.

The correlations between the immunological markers white blood cell count, CD4+ T cell percentage, CD8+ T cell percentage, CD4+ T cell/CD8+ T cell ratio, NK cell percentage, NKT cell percentage, B cell percentage, IL-6, IL-10 and IFN-γ concentrations in serum were evaluated with the Pearson correlation coefficient. The principal component factorial analysis model was used to determine the relationships among variables using the Kaiser–Meyer–Olking measures and the Bartlett sphericity test to validate the results. From the data matrix, explanatory variances were calculated from the principal component model and varimax rotation was used to determine the final correlations. A multivariate general linear model analysis (MLGM or MANOVA) was used for the relationships between immunological markers and epidemiological indicators. All immunological features were included as predictor variables, and age, use of steroids, nutritional diagnosis, and severity of SARS-CoV-2 infection were included as independent variables. The effect size measures were also calculated as a direct indicator of the impact of the epidemiological variables using the partial eta statistic (h2). Moreover, the following values were used for interpretation: h2 < 0.01 as a small or unimportant outcome, h2 between 0.02 and 0.14 as a medium or relevant result, and h2 greater than 0.14 as an important and significant effect. To validate the consistency of the MANOVA models, the Pillai Trace statistic was used to indicate feasibility for calculation. Since this statistic varies between 0 and 1 for values close to unity, it is considered that the MANOVA model can be carried out without problems. A value was considered statistically significant if *p* < 0.05. Data were tabulated with STATA version 17.0 developer StataCorp [17].

## 3. Results

### Characteristics of the Cohort

The sample consisted of 72 pediatric patients whose general characteristics are illustrated in Table 1. Remarkably, 34.7% of the patients were undernourished, and 4.2% were obese–overweight. Around 28% of the individuals had comorbidities: 7 (35%) were asthmatic, 5 (25%) had cancer, 5 (25%) had nephropathy, 2 (10%) had hemoglobinopathy, and 1 (5%) had type 1 diabetes mellitus. Most patients did not have significant medical issues (59.7%). However, 22 patients had MIS-C 22 and 6 bacterial and 1 viral coinfection (Table 1). Moreover, Table 1 illustrates the treatment used; 30 patients did not receive steroid treatment since they did not have critical medical issues during the infection and dexamethasone was often used in the patients that required steroid therapy (30). The prevalent symptom was fever, cough, and diarrhea (Table 1, Figure 1a–c). The distribution of patients by age and clinical severity is depicted in Figure 2.

## 4. The Influence of Age on Lymphocyte Populations and Cytokine Values

Age is an important influence on the hematologic and cytokine parameters observed in the group. Figure 3a illustrates the difference in the leukocyte values obtained in hematologic counts. As expected, the values for the lower age group were higher and decreased with age. The values did not differ between genders.

The levels of cytokines IFNγ and IL-10 are also different with age, as depicted in Figure 3b. No significant differences in gender were recorded. The values of IFN γ increase with age, and the contrary occurs with IL-10. No significant effects with age or gender were recorded with IL-6 values.

## 5. Nutritional Status and Lymphocyte Populations

As recorded in Table 1, the nutritional status of the children admitted to the hospital ward differed independently of the severity or treatment for SARS-CoV-2 infection. In Figure 4a, the values of CD4 significantly decreased in the underweight and overweight individuals (*p* < 0.01). In malnourished individuals compared to overweight–obese patients, the values of CD8 do not compensate for the decrease in CD4 levels. As expected, the CD4/CD8 ratio is higher in the normal and underweight groups and drops dramatically in the overweight–obese group.

Figure 4b depicts that not only the amount of CD3CD4 subpopulation differs according to the nutritional status but also the number of circulating B cells in underweight individuals and NK cells in the obese group as compared to the normal individuals.

## 6. MIS-C, COVID-19 Severity, and IL-6 Concentrations

Higher amounts of serum Il-6 levels were observed in patients with severe disease compared to moderate or mild disease, as shown in Figure 5a. As expected, the values of IL-6 were also significantly higher in patients with MIS-C, Figure 5b.

## 7. Steroid Use, Cytokine Concentration, and Lymphocyte Population

The effect of steroids on the immune response of pediatric cases was analyzed. Statistical differences were recorded in IL-6 and IL-10 serum levels (Figure 6a). The increase in IL-6 in the steroid group refers to the group of MIS-C under treatment. The group of no steroids corresponds with patients with mild and some with moderate disease.

In Figure 6b, the cell populations in the two groups were represented. A significant decrease in the CD3+CD4+ cell subpopulation was observed in the treated group. No other significant differences were encountered.

## 8. Discussion

The immune response against SARS-CoV-2 in pediatric patients differs from adults [2,3,4,6,7,8]. Leukopenia and lymphopenia are frequently reported in adults but are less frequent in the pediatric population [2,3,4,6,7,8]. Moreover, MIS and cytokine storm are generally observed in adult individuals with comorbidities infected with the virus [2,3,4,6,7,8]. There are few studies on pediatric COVID-19, probably because most infected patients have a mild disease course [2,3,4,5,8].

The main target of SARS-CoV-2 is the respiratory tract, and respiratory infections are frequent in the first years of life. These infections lead to memory T and B cell subpopulations that prevent reinfection or severe disease [6,7]. Several factors have been proposed to explain the immune response against pathogens in children [2,3,4,8]. (1) A low expression of ACE2 decreases SARS-CoV-2 infection efficiency in target cells and organs [2,3,4,8,9]. (2) In the early stages of infection, the production of IgM, which has broad reactivity and variable affinity, facilitates the decrease in pathogenic burden [18,19]. (3) In children, fetal-derived B1 lymphocytes contribute to such natural antibody production without antigenic stimulation during the first days of viral infection [18,19]. These B1 lymphocytes and memory B cells are abundant and adaptable to new antigens. Early polyclonal B cell response constitutes an advantage that reduces tissue damage and favors the child’s survival to eliminate unknown or known pathogens [18,19]. In adults, this response is not observed [6]. (4) An efficient innate immune response protects pediatric patients; the cytokine storm is usually observed in adults, not children [2,3,4,6]. (5) A cross-reactive immune response from other coronavirus infections may be protective [4]. Additionally, other common viral respiratory coinfections in children may limit SARS-CoV-2 replication [4,11].

The most common clinical manifestations in pediatric patients with COVID-19 were fever, cough, and diarrhea which have also been reported [2,3,4,20,21,22]. The comorbidities observed in the cohort were asthma (7: 35%) as the most frequent, followed by cancer (5: 25%), nephropathies (5: 25%), hemoglobinopathies (sickle cell anemia) (2: 10%), and type 1 diabetes mellitus (1: 5%). Most patients with comorbidities (67%) had mild disease, suggesting the medical follow-up was successful. Other groups reported no increased risk in asthmatic patients [23,24]. Moderate or severe disease was observed in six (30%) of the patients. This group included four patients with chronic kidney disease on renal replacement therapy (20%) and two cancer patients (10%). 

Several factors were encountered in the cohort analyzed in this report that may affect the response and outcome to the viral infection. The first point is the difference in circulating leukocyte levels and the levels of cytokines present in the serum. The second issue is that a substantial number of patients under five years old were undernourished, suggesting that the course of the disease may be affected [15,25]. It has been described that underweight children have impaired immune response; there is an increase in IL-10 and a decrease in serum IFNγ levels with a reduction in circulating B cells and an inadequate response to polyclonal antigens [25]. We observed a decline in circulating CD3CD4 and B lymphocytes in underweight patients, which agrees with the general description reported without SARS-CoV-2 infection [25]. It is not easy to define the clinical impact of COVID-19 on these patients. To our knowledge, there are no reliable clinical trials to ascertain the role of undernourishment in pediatric patients infected with COVID-19, especially under five years old. 

The prevalence of moderate or severe COVID-19 was higher in children under two years old, suggesting the need to evaluate and provide exhaustive follow-up for SARS-CoV-2 infections in infants. The levels of IL-6 can be an excellent prognostic factor; however, C-reactive protein, the acute phase reactant induced by IL-6, may provide helpful guidelines at a lower cost [26,27]. Among the main differential findings of cytokine levels and cellularity concerning the age group, the highest levels of IL-10 and leukocyte count in infants stand out regarding preschoolers, schoolchildren, and adolescents, which is considered part of the expected physiological behavior due to the maturity of the immune system, where the number of leukocytes is high at birth, with an increase in the first 12 weeks of life and then followed by a decrease and stability during the first two years of age, to then continue a progressive decline in preschool age, school, and adolescence until reaching values similar to those of adults. Breastfeeding in this age group may confer a protective role; breast milk contains many immunologically active components, including interleukin (IL)-10, a major regulatory cytokine of inflammatory responses, which helps infants in the essential immune adaptations [28,29]. The high IL-10 levels may play a role in T reg response in the pediatric population [21,26,27]. The levels of IFNγ only were correlated with age and contrasted with those of IL-10, suggesting that at a younger age, the Th1 response is less prominent. However, in this group, the number of undernourished individuals is higher and possibly nutrition may increase IL-10 and decrease IFNγ production [15,25]. 

The high prevalence of MIS-C is 30.6% [10,11,12,13,14] of the patients, primarily infants and preschoolers (18: 81.8%), who initially presented mild disease. The possibility of overdiagnosis of MIS-C is suggested in this cohort, although IL-6 levels were higher than in the other group [30,31]. In addition, a meta-analysis performed with more than 2000 studies refers to the complexity of the syndrome [32]. Figure 7 represents a mechanism relating to circulating IL-6 levels in MIS-C. 

Differences in circulating lymphocyte populations and cytokine levels were observed in patients similarly treated with steroids, as reported [33,34]. Increased levels of IL-10 and a decreased amount of CD4 were recorded. The values of IL-6 were also increased in the treated group, but some patients were diagnosed with MIS-C, which would probably affect the cytokine levels. The analysis did not include the dose or time of steroids received. 

In a previous report [35], the roles of memory T cells and age were analyzed in adult patients infected with COVID. Younger individuals responded and maintained the amount of memory T cells compared to older individuals whose memory T cells decreased. It would be interesting to assess the memory response of children of this cohort as well as others that have been vaccinated.

The immune response to SARS-CoV-2 infection appears to be a critical factor in the development and prognosis of patients with COVID-19. That is why, by the data presented in this report, it is possible to clarify some hypotheses and open new research topics on the immune response of the pediatric population against SARS-CoV-2 infection. Fortunately, pediatric patients have a lower tendency to develop severe forms and lower mortality. Figure 8 represents a scheme of the findings reported. The differences between infants and preschoolers and teenagers are described.

## 9. Conclusions

No significant alterations in the leukocyte count or percentage of lymphocytes were found with the symptoms. Higher levels of IL-10 were observed in infants. In contrast, higher IFN- γ levels were found in preschoolers, schoolchildren, and adolescents, showing a more dominant antiviral and Th1-type response. The values of IL-6 are related to the severity of the disease, and it is higher in patients with diagnosed MIS-C. The nutritional status and the administration of systemic steroids (dexamethasone use) influenced the levels of cytokines and the percentages of T, and B lymphocytes, and NK cells.

The presence of asthma as a comorbidity was not associated with greater severity. Like recent evidence, it is concluded that it does not constitute a risk factor for contagion or severity of COVID-19 in the Venezuelan pediatric population. It is essential to highlight the differences between eutrophic patients and patients with normal weight and malnutrition in cell populations and subpopulations. These results suggest that nutritional status may be critical in the immune response against SARS-CoV-2.

## 10. Limitations

This study had several restrictions. The availability of COVID-19 pediatric cohort studies (local and regional) to compare our results is limited. The small number of patients with severe disease could partly induce bias in the analysis. The impact of nutrition may also be a bias since it may affect the response to treatment and the progression of the disease. The medical intervention at the ward decreased the effect of malnutrition. 

The cohort was heterogeneous in age, gender, and clinical characteristics.

Since most patients were discharged in a short period due to mild symptoms, patient follow-up was not possible. It is essential to highlight that the clinical approach to the patients was carried out during hospital admission, and treatment was given in the emergency clinic. This early approach favored the low number of patients with severe disease reported. The study did not include the measurement of pulmonary inflammation biomarkers such as plasmatic enolase and acute phase reactants such as C-reactive protein, ferritin, and D-dimer due to economic limitations in processing these parameters in the hospital laboratory. The characterization of the SARS-CoV-2 variants and the clinical correlation were impossible to consider given the limitations in the processing of samples. The samples were transported to another facility.

The ratio of neutrophils/lymphocytes and the ratio of platelets/lymphocytes were analyzed in parallel with the plasmatic concentrations of IL-6, IL-10, and interferon-gamma. However, we could not find any statistical significance or correlation between these parameters, mainly due to the heterogeneity of the cohort.

## Figures and Tables

**Figure 1 biomedicines-11-01409-f001:**
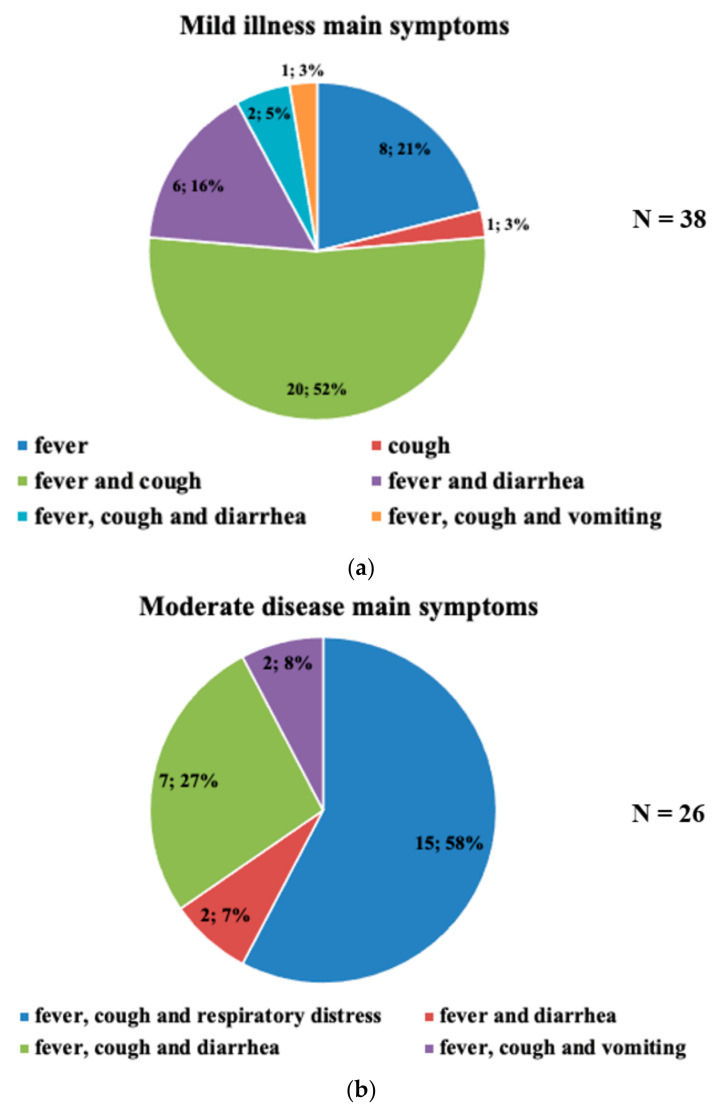
Clinical manifestations according to severity. The sample consisted of 72 pediatric patients. The predominant symptoms were fever, cough, diarrhea, and respiratory distress common in moderate or severe disease patients. (**a**) Symptoms presented in mild disease. The figure shows the distribution of patients according to symptoms and clinical severity. Most patients with mild disease had fever and cough. Microsoft Excel version 16.70 was used to generate the figure. (**b**) Symptoms presented in moderate disease. The figure shows the distribution of patients according to symptoms and clinical severity. Most patients with moderate disease presented with fever, cough, and shortness of breath followed by diarrhea. Microsoft Excel version 16.70 was used to generate the figure. (**c**) Symptoms presented in severe disease. The figure shows the distribution of patients according to symptoms and clinical severity. Most patients with severe disease had a fever, cough, and respiratory distress. Microsoft Excel version 16.70 was used to generate the figure.

**Figure 2 biomedicines-11-01409-f002:**
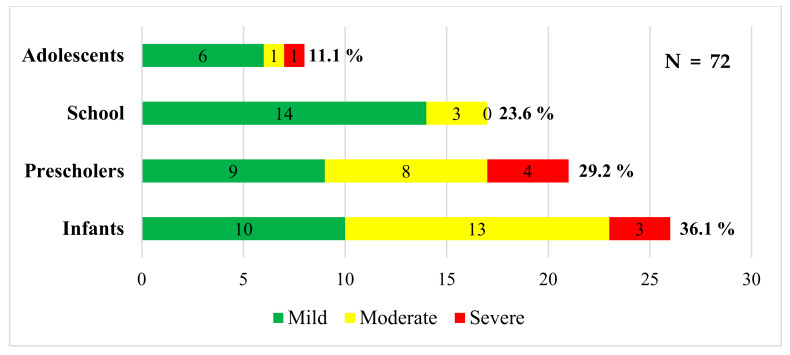
Distribution of patients by age and clinical severity. The figure shows the distribution of patients according to age group and clinical severity. The majority of patients with severe disease were under five years of age. Source: calculations made in Microsoft Excel version 16.70.

**Figure 3 biomedicines-11-01409-f003:**
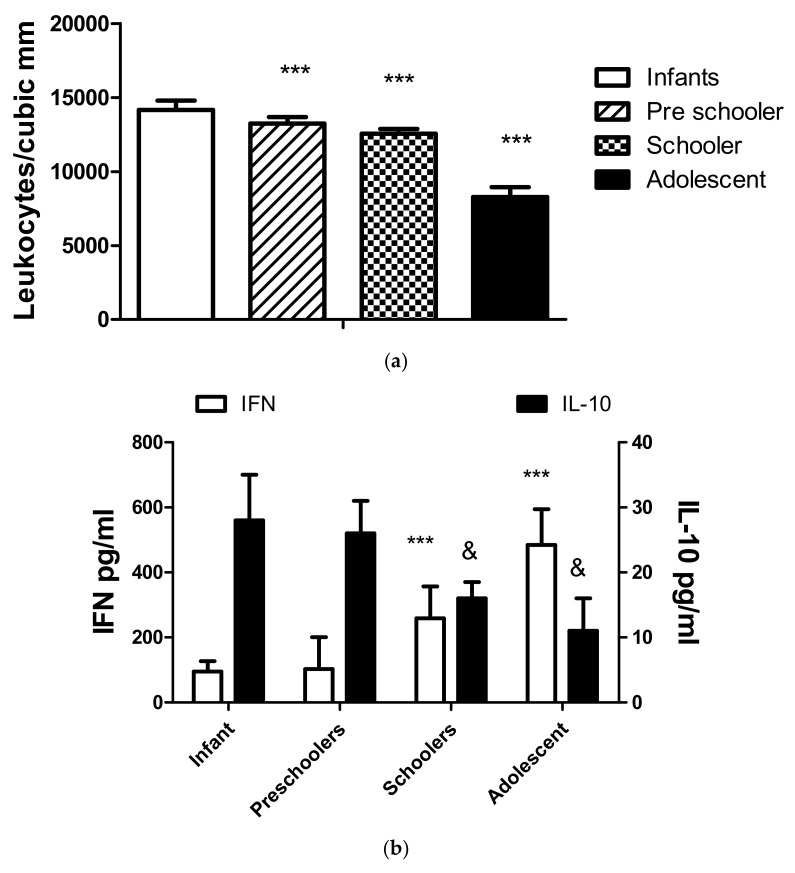
(**a**) Number of circulating leukocytes according to age classification. The figure represents the number of circulating leukocytes in different age groups. The significant differences correspond to comparing the infant and the rest of the groups, *** *p* < 0.0001. The values recorded in adolescents are similar to the normal range in healthy adults. (**b**) Serum cytokine concentration depends on the age group. The left Y-axis and the white bars represent the values of IFN γ. The right axis and the black bars show the values of IL10. The concentrations of IFNγ and IL-10 significantly differ with age when the values recorded in the toddler group were compared to schoolers and adolescents (*** *p* < 0.0001 for IFNγ and & *p* < 0.01 for IL-10).

**Figure 4 biomedicines-11-01409-f004:**
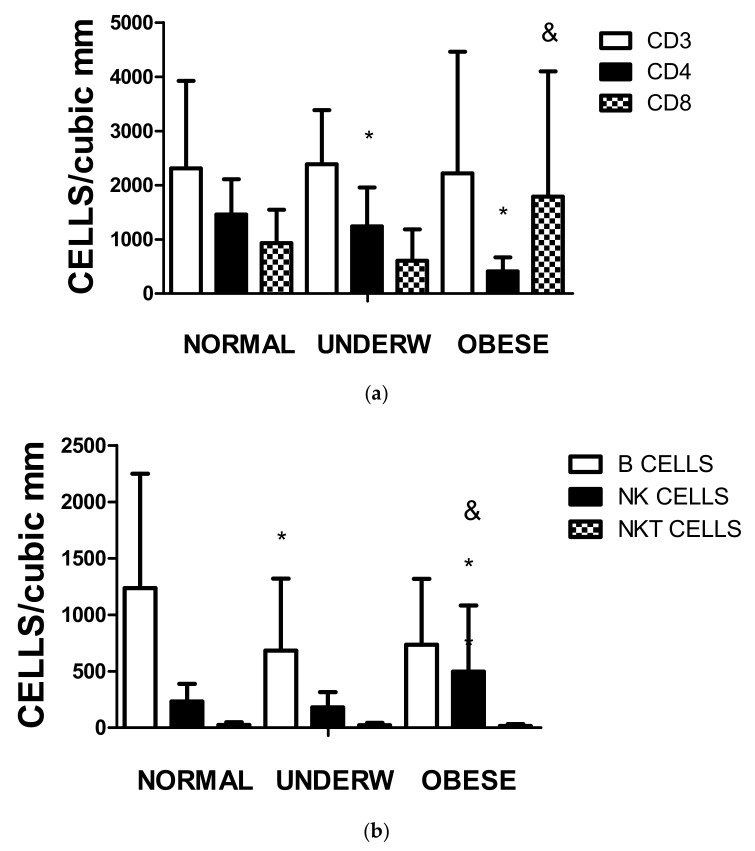
(**a**) The figure illustrates the total amount of T cell and subpopulations according to their nutritional status (Underw refers to underweight and Obese to obese and overweight). The differences among the groups were significant by one-way ANOVA for CD4 and CD8, not for CD3; * representing *p* < 0.01 CD4 values when the normal group are compared to underweight and obese–overweight individuals and & representing *p* < 0.01 CD8 values of the undernourished patients compared with the overweight–obese group. (**b**) The figure illustrates the circulating B, NK, and NKT cell populations (Underw refers to underweight and Obese to obese and overweight). The differences among the groups were significant by one-way ANOVA for B cells and NK cells, not for NKT cells. * representing *p* < 0.01 B cell values when the normal group are compared to underweight, and when NK values of normal individuals were compared with obese–overweight individuals, and & representing *p* < 0.01 NK values of the undernourished patients compared with the overweight–obese group.

**Figure 5 biomedicines-11-01409-f005:**
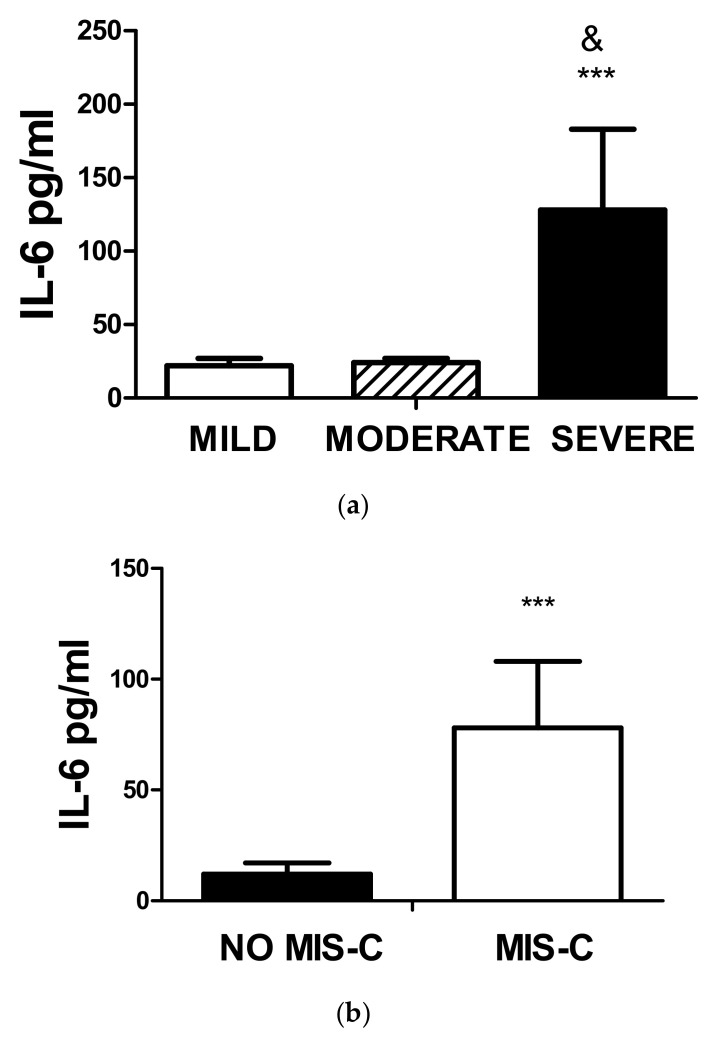
(**a**) Levels of IL-6 and COVID-19 status. The figure represents the values of IL-6 in serum assessed by ELISA of the three groups. Significant differences were observed between mild and moderate (*** *p* < 0.0001) and moderate and severe (& *p* < 0.0001). (**b**) Serum levels of IL-6 and MIS-C. A significantly (*** *p* < 0.0001) higher amount of IL-6 was observed in patients with MIS-C compared to those without it.

**Figure 6 biomedicines-11-01409-f006:**
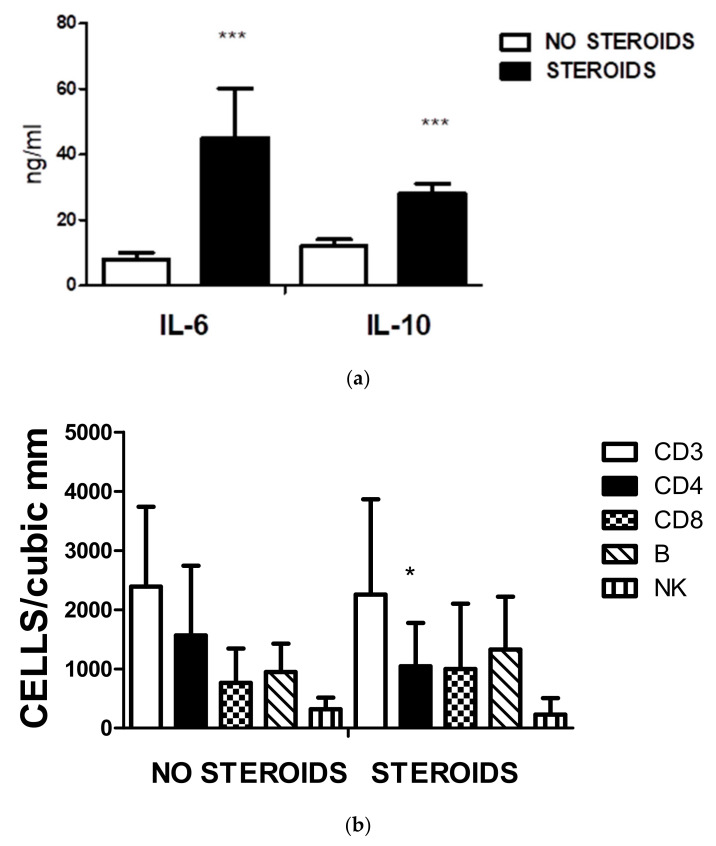
(**a**) Effect of steroids on serum IL-6 and IL-10. The figure illustrates the values of cytokines detected by ELISA in the serum of individuals treated with steroids. The significant differences were recorded using an unpaired Student’s *t*-test (*** represents *p* < 0.0001). (**b**) Effect of steroids on different lymphocyte populations. The only significant (* *p* < 0.01) change in cell number is the amount of CD3CD4 cell subpopulation. There are no other substantial changes in the other cell types.

**Figure 7 biomedicines-11-01409-f007:**
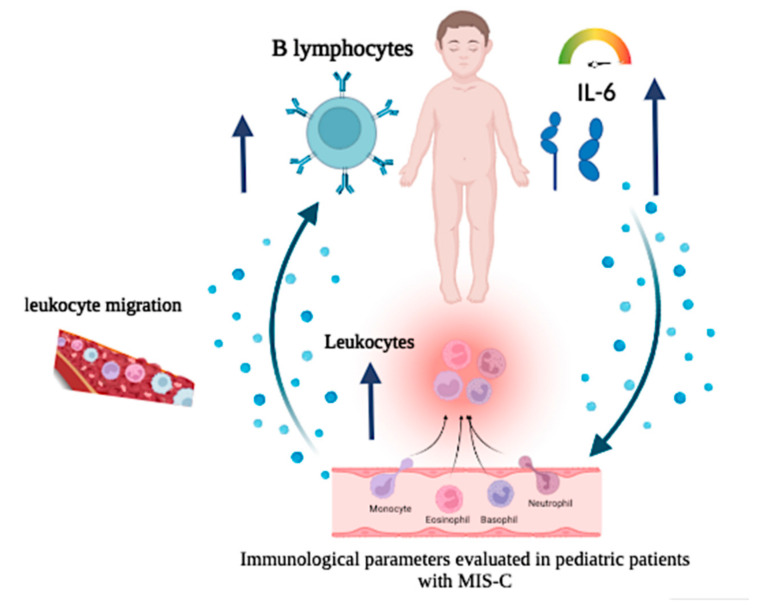
Description: Cellular and cytokine alterations were observed in COVID-19 patients with MIS-C diagnosis.

**Figure 8 biomedicines-11-01409-f008:**
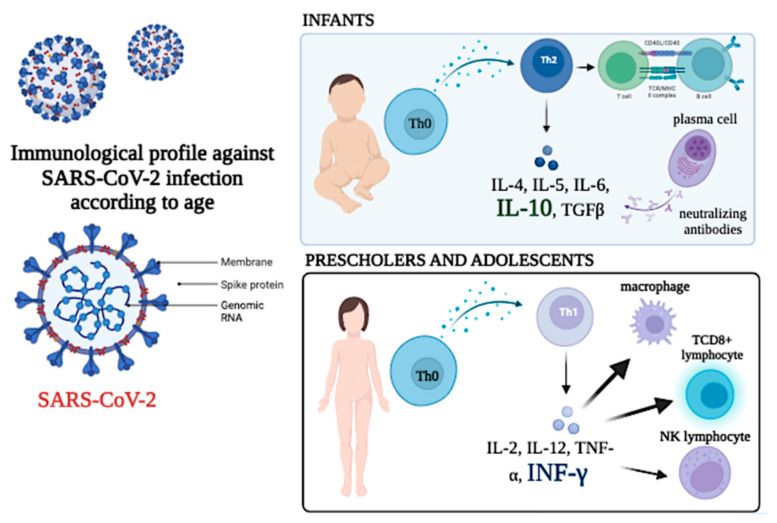
Profile of cytokines and circulating lymphocytes in two different age groups. A Th2 profile in infants and a Th1 profile in schoolchildren and adolescents are suggested.

**Table 1 biomedicines-11-01409-t001:** Characteristic of pediatric patients with SARS-CoV-2 infection.

Variables	*n*	%
Sex		
Male	34	47.2
Female	38	52.8
Age group		
Infant	26	36.1
Preschooler	21	29.2
School-age	17	23.6
Adolescent	8	11.1
COVID-19 severeness		
Mild	38	52.8
Moderate	26	36.1
Severe	8	11.1
Nutrition Diagnosis		
Malnutrition	25	34.7
Normal	44	61.1
Overweight–obese	3	4.2
Comorbidities		
Yes	20	27.8
No	52	72.2
Complications		
No	43	59.7
Bacterial coinfection	6	8.3
Viral coinfection	1	1.4
MIS-C	22	30.6
Treatment received		
No steroids	30	41.7
Dexamethasone	30	41.7
Hydrocortisone	5	6.9
Methylprednisolone	4	5.6
Dexamethasone + immunoglobulin	3	4.2
Symptoms		
Fever	66	91.6
Cough	56	77.7
Diarrhea	38	52.7

## Data Availability

The data are in the Appendix A.

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
