# Peer review of "SARS-CoV-2 Infection in Venezuelan Pediatric Patients—A Single Center Prospective Observational Study"

_biomedicines, 2023, doi:10.3390/biomedicines11051409_

Round 1

Reviewer 1 Report

Dear Authors,

I have read the manuscript and I send you my comments:

1) I agree with the limitations of the study but I think that the power calculation must be added

2) The evaluation and the representation of the results must be changed. I think that Results must be presented considering the difference of Covid-19 symptoms patients and characteristics.

3) Clinical data and drug use must be added

4) Please add the evaluation of plasma enolase, plreviously it has been reported a role of enolase in lung function (PLoS One. 2021; 16(5): e0251819.) 

5) PLease add a gender difference in results obtained

None

Author Response

We would like to thank the reviewer for the useful comments. Please acknowledge that the answer is enclosed in a Word file due to the problem uploading the figures. Thank you

Reviewer 2 Report

This article reviews pediatric cases of COVID-19 in a specialized pediatric hospital in Venezuela .

Few additional information s would be useful :

1° Do the authors have levels of C-reactive protein , ferritin and levels of biomarkers from hemogram (neutrophil to lymphocyte ratio and platelet to lymphocyte ratio) in these children and how do they parallel IL6 , IL10 and interferon gamma.

2° Did the authors caracterize the variants of SARS-CoV2 and were variants associated with peculiar clinical characteristics ?

Globally , English language is of quality ; a final review by an English-speaking native could add value to the article

Author Response

We would like to thank the reviwer for the useful discussion. We are adding a Word file with the answers since it was difficult to upload the figures.

Round 2

Reviewer 1 Report

Dear Authors,

I have read the manuscript and I inform you that I have not other comments. 

none